# Polydopamine-Based Material and Their Potential in Head and Neck Cancer Therapy—Current State of Knowledge

**DOI:** 10.3390/ijms24054890

**Published:** 2023-03-03

**Authors:** Marta Witkowska, Ewelina Golusińska-Kardach, Wojciech Golusiński, Ewa Florek

**Affiliations:** 1Faculty of Chemistry, Adam Mickiewicz University, 61-614 Poznan, Poland; 2Centre for Advanced Technologies, Adam Mickiewicz University, 61-614 Poznan, Poland; 3Department and Clinic of Dental Surgery, Periodontal Diseases and Oral Mucosa, Poznan University of Medical Sciences, 60-812 Poznan, Poland; 4Department and Clinic of Head and Neck Surgery and Laryngological Oncology, Poznan University of Medical Sciences, 61-866 Poznan, Poland; 5Laboratory of Environmental Research, Department of Toxicology, Poznan University of Medical Sciences, 60-631 Poznan, Poland

**Keywords:** polydopamine, biomaterials, head and neck, cancer therapy

## Abstract

Head and neck cancers (HNC) are among the most common cancers in the world. In terms of frequency of occurrence in the world, HNC ranks sixth. However, the problem of modern oncology is the low specificity of the therapies used, which is why most of the currently used chemotherapeutic agents have a systemic effect. The use of nanomaterials could overcome the limitations of traditional therapies. Researchers are increasingly using polydopamine (PDA) in nanotherapeutic systems for HNC due to its unique properties. PDA has found applications in chemotherapy, photothermal therapy, targeted therapy, and combination therapies that facilitate better carrier control for the effective reduction of cancer cells than individual therapies. The purpose of this review was to present the current knowledge on the potential use of polydopamine in head and neck cancer research.

## 1. Introduction

Head and neck cancers (HNC) are very common on a global scale. In 2018, they were in sixth place in terms of the most common cancers in the world [1]. According to The Surveillance, Epidemiology, and End Results (SEER) program HNC in 2022 it is estimated that there will be 54,000 new cases of the oral cavity and pharynx cancer and an estimated 11,230 people will die of this disease. Oral cancer is more common in men than women, among those with a history of tobacco or alcohol use, and people infected with human papillomavirus (HPV) [2] (Figure 1). Smoking can cause head and neck cancer and is correlated with the frequency and intensity of smoking [3]. Another factor is the consumption of alcohol, which acts as a solvent, increasing the exposure of the mucous membranes to carcinogens [4]. The risk is significantly increased when smoking and drinking alcohol at the same time. However, individual variability in genetic susceptibility plays an important role because not all smokers and drinkers develop HNC [5]. Head and neck tumors constitute a large group of neoplasms and include organs such as lips, oral cavity, pharynx, paranasal sinuses, larynx, and ear [6]. One of the most common is squamous cell tumors, which are moderately sensitive to radiation and chemotherapy. In approximately 30–40% of patients, HNC is present in the early stages of the disease and has a 5-year survival of 70–90% with treatment [7]. Most cases of HNC are diagnosed in the advanced stages when medical treatment is less effective and surgical treatment cripples organs necessary for speech and swallowing [7].

HNC is a complex and difficult disease. In the early stages, the primary treatments include surgery, radiation therapy, chemotherapy, immunotherapy, gene therapy, photothermal therapy (PTT), and photodynamic therapy (PDT) [1]. Surgery is the primary treatment of oral cancer, while radiotherapy is the primary treatment for nasopharyngeal cancer. Due to the anatomical sensitivity of these tumors and surrounding tissues, current treatments may result in adverse effects such as mucositis, neurotoxicity, tissue or bone necrosis, fibrosis, and even infection [2]. The Food and Drug Administration (FDA) has approved various chemotherapeutic agents such as cisplatin, carboplatin, 5-fluorouracil, docetaxel, methotrexate and bleomycin, and three monoclonal antibodies for the treatment of HNC. The current standard of treatment for recurrent head and neck cancers focuses on chemotherapy based on cetuximab and platinum with cisplatin or carboplatin as well as methotrexate and 5-fluorouracil, and doctors additionally introduce surgery and radiotherapy [3]. Chemotherapy diffuses its distribution, which reduces the effectiveness of the treatment and leads to severe side effects. Radiotherapy, in turn, often promotes the development of tumor resistance, leading to a negative prognosis [4]. It is important to use multidisciplinary treatment in the therapeutic process, as well as to adjust the treatment plan continuously according to changes in the patient’s body [5]. Patients with relapsed HNC and disseminated metastases do not respond to treatments, such as surgical ablation in combination with radiotherapy and chemotherapy. Resistance to cancer metastasis is largely due to the different and heterogeneous subpopulations of metastatic cells, in which they modify gene expression, growth rate, properties, and cell surface functions compared to primary cancer cells, which may be the cause of resistance to commonly used drugs and a problem with drugs reaching metastatic sites, contributes to poor therapy outcomes [6]. 

The arising applications of nanotechnologies in biomedicine provide new opportunities for dealing with the problems of therapies in cancer treatments. Nanomaterials are particles at the nanometric scale that have great potential in the medical field due to their specific material properties [7]. Nanomedicine may have a tremendous impact on head and neck cancer therapies through its targeted approach and potential reduction of side effects. The controlled delivery of drugs is very important to the pharmaceutical industry because such therapy allows for the delivery of a higher concentration of the drug to the tumor cells while giving patients a lower dose. In the case of HNC treatment, it is very important due to the lack of specificity of conventional cytotoxicity agents [8]. Nanocarriers smaller than 100 nm may be a vehicle for systemic administration due to their prolonged blood circulation [9]. The induction of cytotoxicity in neoplastic cells depends on the size of polydopamine nanomaterials (PDA). Due to their small size, nanoparticles can be captured by cancer cells through the effect of increased permeability and retention (EPR), causing local accumulation and cytotoxic effects on these cells [10]. Polydopamine can be easily synthesized by simple dopamine oxidative self-polymerization and due to its excellent biocompatibility, degradability, low toxicity, and good photothermal conversion efficiency, it can serve as an ideal nanocarrier or photothermal agent for cancer treatment. 

Importantly, due to its excellent photothermal effects and strong adhesive capacity, PDA can be easily functionalized with numerous nanomaterials for synergistic anti-cancer therapy [11]. Herein, we describe the current status of various polydopamine-based nanostructures administered to support the treatment of HNC and describe the potential future use of polydopamine.

## 2. Synthesis an Characterisation of Polydopamine

Curiosity to discover how mussels adhere to various wet surfaces with a force that can withstand ocean currents led to the discovery of adhesive proteins secreted by mussels, which inspired the creation of the compound PDA, which turned out to be crucial for mollusks adhesion [12]. Polydopamine is a brown-black, insoluble biopolymer [13]. It is formed in the process of dopamine oxidation in alkaline conditions [14]. PDA is composed of indole units with varying degrees of hydrogenation, probably linked by carbon-carbon (C-C) bonds between the benzene rings. The polymer has the ability to tautomerize quinoid and catechol units, which results from the presence of two oxygen atoms in the structure bound to the benzene ring. Polydopamine is also made of dopamine units that are not cyclic, i.e., they contain aminoethyl side chains [15]. An undoubted advantage of PDA is also the ease of its preparation—dopamine undergoes self-polymerization in mild conditions [16]. PDA synthesis is described as fast and simple, and the required reagents as inexpensive [17]. The monomer of polydopamine is dopamine. There are several ways to obtain polydopamine: by oxidation in an aqueous solution, by electropolymerization and by enzymatic oxidation [18]. The first method consists in dissolving dopamine hydrochloride in an alkaline solution. In the presence of oxygen, spontaneous auto polymerization of dopamine to polydopamine occurs. Liu et al. observed a color change of the solution from colorless through pale yellow to dark brown during the reaction [19]. For example, Sahiner et al.’s team synthesized PDA in Tris buffer at pH 8.5 at room temperature with 300 rpm agitation. After 24 h, precipitated PDA particles were observed [20]. The advantage of this method is its simplicity, no need to use environmentally harmful reagents, complicated equipment or to ensure extreme reaction conditions [19]. The thickness of the film can be adjusted by changing the concentration of dopamine and appropriately extending or shortening the polymerization time. The disadvantage of the above method is the difficulty in obtaining a film layer of uniform thickness. The electropolymerization method produces a film of greater and more uniform thickness and the obtained polydopamine can be deposited directly on electrodes. The reaction takes place in the absence of oxygen, however he limitation of the method is the fact that the PDA film can only be created on electrically conductive materials. The third method of obtaining polydopamine is enzymatic oxidation, in which the enzyme tyrosinase catalyzed the dopamine oxidation reaction [18].

## 3. Properties of Polydopamine Used in the Treatment of Cancer

Scientists paid attention to the excellent adhesive properties of polydopamine, which enables it to be deposited on the surface of all types of organic and inorganic substrates, even highly hydrophobic ones, creating a stable coating [19,20]. Polydopamine is a hydrophilic compound. This provides the desired properties of the surface to be coated with the compound without the need for other modifications. As a result, polydopamine can be used as a substance that improves hydrophilicity [21]. Their polar groups are responsible for the high surface energy and hydrophilicity of the polydopamine molecule [22]. As the main pigment of naturally occurring melanin, PDA has excellent optical properties and good biocompatibility [20]. Other advantages of polydopamine, deciding its wide application, including in medicine, there is sensitivity to changes in pH and biodegradability [23]. Polydopamine also can absorb near-infrared radiation and convert it into heat with high photothermal conversion, which makes it possible to use this nanomaterial in photodynamic therapy [24]. Eumelanin and polydopamine have close absorption spectra in the UV-VIS wavelength range in the electromagnetic spectrum. Their quantum fluorescence efficiency is low. Both of these compounds have the ability and can transform most of the absorbed light into heat [25]. Liu et al. showed that irradiation with a near-infrared laser of 2 W/cm^2^ near-infrareds, a suspension of polydopamine nanoparticles, can increase the temperature by about 33.6 °C, compared to a water sample where the temperature increased by 3.2 °C. The scientists found that neoplastic tissues, after injection of polydopamine and irradiation with a laser, can be heated to a temperature of 50 °C, which would result in the death of the cancer cells after 5 min. The high thermal conversion efficiency of polydopamine allows it to be used as a photothermal material [26]. They can use the polymer in photothermal therapy for cancer treatment, which is characterized by high selectivity and low invasiveness, and near-infrared laser for a specific tumor site. Therefore, polydopamine is a promising candidate for anti-cancer therapy [25]. PDA molecules are rich in reducing functional groups—as a result, polydopamine shows a wonderful ability to scavenge free radicals and, reactive oxygen species (ROS), and thus reduce inflammation caused by ROS [27]. Due to its chemical structure, polydopamine can be easily modified by a reaction between amino or thiol groups and quinone groups present in the structure of polydopamine [28]. Another property of polydopamine resulting from its specific structure is its high ability to bind and transport drugs (e.g., doxorubicin)—through π-π bonds or hydrogen bonds. At the same time, a constant pH-dependent release of the transported particles was observed [28]. Cui and colleagues conducted a study that confirmed the pH-dependent release of doxorubicin from the PDA nanocarrier. Drug release was analyzed in buffer at a pH ranging from 5.0 to 7.4. It was observed that doxorubicin was released slowly at pH 7.4, while at pH 6.0 this rate increased, and at pH 5.0 almost 85% of the drug was released within 12 h [29]. Because the tumor microenvironment is acidic [30], this mechanism may be of particular importance in the design of the controlled release of drugs based on nanomaterials. Polydopamine can chelate metals with oxygen or nitrogen atoms of the molecule, which is dependent on pH, which makes it possible to complex polyvalent metal ions, e.g., iron (III), copper (II), and zinc (II) ions. This enables binding with radioisotopes and transition metals, which makes it possible to use polydopamine in radioisotope therapy for cancer or to purify water from heavy metals [31,32]. Reactions of polydopamine with coated surfaces allow for many modifications of these surfaces, which enable obtaining the desired properties. Zmerli et al. synthesized polyethylene glycol (PEG) modified polydopamine nanoparticles to combine photothermic therapy and photodynamic therapy, thereby increasing the effectiveness of anti-cancer therapy [33].

## 4. Targeted Therapy of Polydopamine-Based Nanomaterials

Targeted therapy is a type of precise treatment, with targets proteins that control cancer cells’ growth and spreading throughout the body. Neoplastic tissue accumulates nanoparticles faster than other tissues. This is the so-called enhanced permeability and retention effect, which is responsible for passive targeting (Figure 2). Active targeting can be achieved through molecular recognition, which enables the homogeneous distribution of nanoparticles in the tumor tissue and thus delivery of the drug to a specific site [34]. Appropriate modification of the surface of nanoparticles can extend their circulation time in the blood and also reduce the uptake of macrophages by the phagocytic system. Thus, nanoparticles can increase the selectivity and effectiveness of methods of provoking the death of cancer cells with minimal toxicity to non-malignant cells [35]. An important reason for nanoparticle modification is the control of nanoparticle interactions with cells, enabling the targeting of peptides, ligands, or small particles [36]. One example is the epidermal growth factor receptor (EGFR), which overexpression is common with head and neck squamous cell carcinoma (HNSCC). Its high level is associated with poor prognosis in various neoplasms, which indicates the usefulness of receptor-targeted therapies. Another one is folic acid (FA) receptors, which are attached to the surface of cells with a high affinity for folic acid and are overexpressed in many malignancies such as breast, ovarian, lung, kidney, head and neck cancer, etc. [37]. Folic acid is a water-soluble B vitamin that is essential for DNA synthesis. Folic acid retains the ability to bind to folate receptors after being coupled to other structures and often improves endocytosis [11]. Properties of polydopamine enable them to penetrate through the cell membrane, including the blood-brain barrier, increase the half-life of active substances, or delay their metabolism.

Biomaterials for more effective drug delivery at HNC are strongly associated with unique features of the tumor microenvironment, such as low pH, high ROS levels, enzyme overexpression, and hypoxia [38]. Targeted drug delivery nanomaterials respond to these changes in the microenvironment, leading to more accurate drug delivery, better tumor penetration, and sustained drug release, while reducing the dose of chemotherapeutic agents delivered, resulting in fewer side effects [39]. Polydopamine-based nanomaterials can be used to target therapy due to their pH dependence. Drug release may be based on the effect that the pH around the tumor is slightly acidic compared to healthy tissues [40]. Studies by Cheng et al. showed that polydopamine-modified mesoporous silica nanoparticles loaded with doxorubicin released smaller amounts of a chemotherapeutic agent than other nanoparticles at pH = 7.4. The release of the drug polydopamine was significantly higher at lower pH values, especially at pH = 5.0. Polydopamine had a positive effect on the release of doxorubicin at the tumor site, where there is an acidic environment [41]. Most tumor cells, even under normal oxygen conditions, obtain energy mostly through glycolysis. The ongoing process of glycolysis and the resulting product—lactic acid, lowers the pH to about 6.0, which differs from the pH of healthy tissues, which is around 7.2–7.4. Acidification of the environment significantly accelerates the delivery of the drug to the cancer cells [42]. Other aspects in the tumor microenvironment which may lead to the targeted release of the drug are high-level or reactive oxygen species and occurring glutathione. On the surface of PDA, there are reactive phenolic or hydroxyl groups that are oxidized by hydrogen peroxide. As a result, the hydrogen bonds between the drug and polydopamine are weakened. Glutathione can apply the same effect by disrupting the π-electron bonds [42]. A high concentration of glutathione makes it possible to apply therapy based on targeted drug delivery and release, depending on the level of glutathione (GSH) [27]. The connections of disulfide bond break and release the drug when exposed to an environment with a lot of GSH [43]. Drug delivery systems are modified to achieve the highest selectivity. It is possible through biological pathways and targeting of polydopamine, for example, individual receptors, peptides, and even genes [44]. One of the methods of selective therapy is to take advantage of the fact that tumors overexpress folic or hyaluronic acid receptors. Coating the polydopamine surface with analogous ligands improves the targeting of the therapy and ensures its effectiveness. Modifying with peptides allows binding and, as a result, contributes to increasing the impact of polydopamine on cells [40]. 

In vivo application of interfering strategy via siRNA or shRNA uprise many challenges, like transfection and tissue targeting. Due to the excessive hydrophilic properties of these molecules, and poor in vivo stability, there is a need to use the vectors [45]. Modified mesoporous polydopamine (MPDA) nanoparticles proved useful for delivering siRNA in thyroid cancer cells. In their research Martimprey et al. and Niemela et al., 2020 showed, that nanoparticles were able to accurately and effectively deliver siRNA or drugs into thyroid cancer cells [45,46]. Asghar et al. 2021 prepared mesoporous polydopamine particles with surface modification by N,N-dimethylethylenediamine (DMEA) and loaded them with siRNA. The cells efficiently took up these nanoparticles and released the siRNA after 96 h. They were also investigating the effect of empty or STIM1-siRNA loaded nanoparticles on cell invasion and proliferation, however, they found no effect of empty nanoparticles, and both the invasion and proliferation were significantly downregulated by STIM1-siRNA loaded nanoparticles [47]. Since the use of pharmacological blockers or siRNA in the inhibition of e.g., STIM1 is problematic, Asghar et al., showed that siRNA-loaded nanoparticles could be used to successfully knock down STIM1, and thereby reduced both proliferation and invasion of the ML-1 cells. A combination of siRNA-loaded nanoparticles and lower doses of cytostatic could be an alternative approach to critical thyroid cancer progression, causing less adverse effects of the chemotherapy [47].

## 5. Polydopamine in Chemotherapy

Chemotherapy is the most widely used cancer therapy. Doctors recommend combined comprehensive multimodal treatment for patients with terminal-stage of head and neck cancer [21]. We can divide the use of chemotherapy into induction chemotherapy, synchronous chemotherapy, and adjuvant chemotherapy [22]. Induction chemotherapy is used before surgery to reduce the volume of tumors, synchronous chemotherapy is used to increase the effectiveness of radiotherapy while reducing the risk of lymph node metastases. The goal of adjuvant chemotherapy, on the other hand, is to kill small lesions that cannot be surgically removed or to reduce relapses and improve survival. In chemotherapy over HNC treatment, drugs such as fluorouracil (5-FU), methotrexate (MTX), bleomycin, mitomycin C, hydroxyurea, cisplatin, and carboplatin are used [4]. Chemotherapy is very effective in fighting against both primary tumors and metastases, however, it has its drawbacks. One of them is low selectivity—chemotherapeutic agents do not limit their cytotoxic activity only to neoplastic cells, but they also destroy healthy cells and tissues [23]. Another issue is the growth of resistance to these drugs. In patients with advanced neoplastic disease, cytostatic resistance is one of the most serious therapeutic challenges [24]. Controlled drug delivery systems are one way to increase the effectiveness of chemotherapy and decrease the risk of side effects. Nanocarriers can easily deliver drugs directly to the tumor, taking advantage of the phenomenon of increased blood vessel permeability at its site. As a result, bioavailability increases, while the toxic effect of drugs on healthy cells and the risk of resistance occurrence are reduced. The consequence is also a decrease in the costs of therapy [23]. Cheng et al. designed a folding nanocarrier system of d-a-tocopheryl polyethylene glycol 1000 succinate (TPGS)-functionalized polydopamine-coated mesoporous silica nanoparticles for the delivery of cytostatics—doxorubicin (DOX). The effect of the obtained nanocomposite (MSNs-DOX@PDA-TPGS) on drug-resistant lung cancer cells was investigated in vitro and in vivo. MSNs-DOX@PDA-TPGS showed greater toxicity to multi-drug resistant cancer cells than free DOX. It was also observed that MSNs-DOX@PDA-TPGS can increase the concentration and extend the residence time of DOX at the tumor site. The increased release of DOX in the tumor’s acid microcirculation, and thus the lower toxicity of the drug in healthy cells, was due to the sensitivity of PDA to changes in pH [25]. Polydopamine systems with doxorubicin are often used [11]. In the studies of Wang et al., the combination of doxorubicin and gossypol gives beneficial effects resulting from synergistic action. However, the π-π bonds between them are not stable enough. This may be due to the different hydrophilic nature of the two substances. The use of polydopamine, which also contains π electrons in its structure, increases the stability of the system used. This results in an extension of the biological half-life, and this translates into the effectiveness of the therapy [23].

## 6. Polydopamine in Radiation Therapy

Radiotherapy is a widely used treatment that adopts high-energy rays to inhibit the proliferation of cancer cells. However, the high-energy rays inevitably induce damage to the normal tissues, exerting hazardous effects related to radiotherapy [26]. One of the methods to improve is radiosensitization, introducing agents, that would make the tumor more sensitive to ionizing radiation [27]. Nanocarriers such as polydopamine can transport not only drugs but also radioisotopes and thus increase the selectivity and effectiveness of radiation therapy, which was the subject of a study by Zhong et al. [28]. The polydopamine synthesized by them has been functionalized with polyethylene glycol to increase its stability. Then, the technetium 99mTc radioisotope was introduced into some of the particles, and the 131I radioisotope and doxorubicin were introduced into some of the particles, resulting in the 131I-PDA-PEG/DOX nanocomposite. The effectiveness of the resulting complex in chemotherapy combined with radiotherapy has been studied both in vitro and in vivo. In vitro, it was observed that the cytotoxicity of 131I-PDA-PEG was higher than the cytotoxicity of 131I alone—the use of a carrier significantly increased the uptake of radioactive iodine by tumor cells. In contrast, the toxicity of free doxorubicin and PDA-PEG/DOX was comparable. In an in vivo study, mice that received combination therapy with 131I-PDA-PEG/DOX showed the highest level of tumor cell destruction. Importantly, no significant long-term toxicity was observed [28].

## 7. Polydopamine in Photodynamic Therapy

Photodynamic therapy involves the non-invasive introduction of a drug into the tumor cell to destroy it [29]. Clinical studies have shown that during PDT, the oral cavity of oral squamous cell carcinoma (OSCC) patients retains its structure, function, and appearance as well as minimizes side effects without permanent or systemic toxicity [30]. It is a method that can be a good alternative treatment for patients who are difficult to treat with surgery [31]. Due to its favorable properties, polydopamine can be used as a carrier. Targeted delivery to the tumor can be ensured, which reduces and spares the accumulation of the active ingredient in healthy cells. Most conventional photosensitizers (PS) are stimulated by visible light (VIS), which cannot reach deeply located tumor tissues. Therefore, near-infrared (NIR) light has begun to be used in PDT, as it can penetrate deep into tissues. Studies have shown that nanoparticles can convert NIR light into VIS, which can then be absorbed by PS [32]. PDT uses photosensitizers which, under the influence of irradiation, pass from the ground state to the excited state, i.e., the triplet one [33]. In its active state, the substance molecule transfers energy to oxygen and stimulates the production of singlet oxygen. The resulting product may further generate free radicals. A significant increase in their level causes the death of the cancer cell. Most of the substances used are hydrophobic, so the advantage of polydopamine, which is increasing the hydrophilicity of the system, is used, which translates into more efficient delivery to the target [34]. Photosensitizers can be bound by PDA through chemical interactions. The attachment of substances takes place due to the presence of amine or thiol groups. Also, the presence of carboxyl groups allows the carbodiimide reaction to take place and the bond to be formed. In addition to chemical bonds, physical processes can also take place. Compounds containing aromatic groups react through π electrons [33]. The photosensitizers used include, among others porphyrins, or macrocyclic aromatic systems. They are characterized by strong fluorescence and the ability to generate free radicals. PDA increases the hydrophilicity of these systems for effective delivery and therapy. Apart from porphyrins, various metal complexes are also used, e.g., ruthenium, iridium, and gold [35]. In the study by Yan et al., A PDA carrier modified with the presence of folic acid (FA) was created (folate receptors (FR) are present in the neoplastic tissue in much greater amounts than in the healthy tissue—this feature can be used as a therapeutic and diagnostic target). After loading the cationic phthalocyanine (Pc) photosensitizer, the PDA-FA-Pc nananolide was obtained, the effect of which was tested in vitro and in vivo (mouse model). The photosensitizer was gradually released in the acidic environment of the tumor. Much higher PDA-FA-Pc uptake by cancer cells than by healthy cells was observed. Finally, the investigated PDA-FA-Pc nanocomposite caused a significant inhibition of tumor growth [36].

## 8. Polydopamine in Photothermal Therapy

The purpose of photothermal therapy is to increase the temperature in tumor tissues while preventing damage to surrounding healthy tissues [37]. Local hyperthermia can have direct cell-killing effects. Nanoparticle-mediated photothermic therapy uses photosensitive light-to-heat conversion to effectively remove neoplastic tissue, and due to the limited depth of light penetration, it is more suitable for the treatment of superficial tumors such as skin cancer and HNSCC [38]. In the case of single cancer cells, the heating process may change the permeability of the cell membrane and receptors, change the enzymatic activity and cell structure, inducing apoptosis of single cancer cells. At the same time, exposure of cells to heating causes rapid translocation of nucleolin from the nucleolus into the nucleoplasm, which inhibits DNA replication and synthesis. The main factors in the tumor microenvironment that have a strong influence on the tumor response to hyperthermia are perfusion, permeability, pO2, pH, and pressure [8] (Figure 3). There is a need for photothermal converting agents (PTAs) with high tumor accumulation and photothermal conversion performance [39]. Due to the slow or non-degradable photo thermal converting agents based on metals such as gold nanopeel, gold nanocoating, gold nanocage, and CuSx nanocrystal, they are retained in many organs after the mission is completed. It was found that in the case of hollow gold nanospheres, about 70 and 95% of them can be retained in the liver and spleen, respectively [40].

A special property of polydopamine is its photothermal conversion ability [23]. Under the influence of light with a wavelength of λ = 808 nm, the emission of heat by polydopamine is most effective. Also, other ranges may cause this phenomenon, e.g., below 1064 nm, which results from differences in the thickness of the PDA layer [41]. Polydopamine can occur in various shapes or in the packing densities of oligomers, which means that it can have different photothermal properties, depending on its properties. For example, nanoparticles that are less than 200 nm are more commonly used in in vivo research because of their pharmacokinetic properties. Temperature changes in photothermal therapy show a dependence on the concentration of nanoparticles, but in the case of high PDA concentrations, they have a greater light attenuation factor, and thus a smaller penetration depth [42]. In this therapy, photosensitizers are used, most often chlorin e6, which absorbs photons from the active radiation. This allows oxygen to move from a singlet, ground to an excited state. The produced products react with other particles, transferring energy to them, and also contributing to the generation of further free radicals [34]. Upon radiation exposure, conversion takes place and heat is released. This has the effect of hyperthermia and this contributes to the induction of apoptosis. The mechanism causing cancer cell death is related to the induction of oxidative stress in mitochondria under the influence of temperature [43]. The pathways associated with the pro-apoptotic protein Bid, cytochrome c, and caspase 3 are activated [44]. The mechanisms related to the generation of ROS occur at the temperature of 41–45 °C. The uptake of drugs delivered in the polydopamine system by cancer cells also increases to this extent. On the other hand, increasing the temperature to 46 °C causes necrotic death [44]. In photothermal therapy and photodynamic therapy, the ability of polydopamine to affect reduced glutathione is used. PDA reacts with GSH and reduces its level in the cancer cell environment [45]. This contributes to the weakening of the antioxidant effect of the tripeptide. As a result, the level of free radicals generated when using PTT or PDA is higher, and the applied therapy is more effective [46]. However, the limited area of action of PPT carries the risk of recurrence of the disease, as hidden neoplastic cells may remain around the tumor beyond the radiation range. Hence, combining PTT with other types of cancer therapy seems beneficial. The team of Chen et al. developed a combination of photothermal therapy with immunotherapy and chemotherapy. As a carrier for doxorubicin and imiquimod, nanoparticles of polydopamine inoculated with folic acid, a ligand with a high affinity for folic receptors abundant in neoplastic tissue were used. The obtained particles had several unique advantages, including ease of manufacture, full biocompatibility, and high drug delivery efficiency to the tumor. The combination of the above forms of therapy gave a synergistic effect in the fight against cancer—PTT and chemotherapy led to the almost destruction of the tumor, and immunotherapy saved the mice participating in the experiment from tumor recurrence [47].

## 9. Polydopamine in Combined Therapy

Photothermal therapy is one of the non-invasive cancer treatments. It is based on the conversion of photon energy into heat energy, which is cytotoxic to cancer cells. PTT is a local therapy with relatively few side effects [48]. However, the limited area of action of PPT carries the risk of recurrence of the disease, as hidden neoplastic cells may remain around the tumor beyond the radiation range. Hence, combining PTT with other types of cancer therapy seems beneficial. The team of Chen et al. developed a combination of photothermal therapy with immunotherapy and chemotherapy. As a carrier for doxorubicin and imiquimod, nanoparticles of polydopamine inoculated with folic acid, which is a ligand with a high affinity for folic receptors abundant in neoplastic tissue, were used. The obtained particles had several unique advantages, including ease of manufacture, full biocompatibility, and high drug delivery efficiency to the tumor. The combination of the above forms of therapy gave a synergistic effect in the fight against cancer—PTT and chemotherapy led to the almost destruction of the tumor, and immunotherapy saved the mice participating in the experiment from tumor recurrence [47].

Polydopamine is also used in gene therapy [49]. Its purpose is different from the case of chemotherapy or radiotherapy. The supply of nucleic acids does not act on the symptoms but on the source of the disease, which are various mutations that promote tumorigenesis. However, DNA is sensitive to degradation by lysosomes. Due to its properties, polydopamine can be used as a carrier. In this case, PDA with a modified polyethyleneimine [50] was used. The ability to photochemically convert under the influence of radiation is also advantageous. The generated heat and free radicals contribute to the disruption of endosome membranes, which translates into reduced nucleic acid uptake. By passing the fundamental barrier that interferes with drug delivery, it is possible to increase the effectiveness of the therapy [11,51]. In a study by Zhang et al., polydopamine was grafted with folic acid, creating a nanocarrier for siRNA—a short interfering RNA that can silence the expression of sequences homologous genes—in this case, the ROC1 oncogene [Zhang et al., 2021]. Gene silencing by RNA interference appears to be a promising therapeutic strategy. To improve the safety, biodistribution, pharmacokinetics, selectivity, and efficacy of siRNA therapy, it is necessary to use appropriate nanocarriers [52]. The nanovector designed by Zhang and colleagues was characterized by good biocompatibility and, thanks to the presence of FA, selectively delivered the transported siRNA directly to liver cancer cells through receptor-associated endocytosis. Then, siRNA was released from the nanocomposite into the tumor microenvironment—this process was conditioned by the change in pH. As a result, not only was the proliferation of neoplastic cells inhibited but also their apoptosis was stimulated. The combination of the above gene therapy with photothermal therapy has shown an excellent inhibitory effect on the growth of liver cancer, in vitro and in vivo [53].

## 10. Polydopamine Used in Head and Neck Cancer Therapy

Nanomedicine is a rapidly developing field. Due to the small size of nanocarriers, smaller than 100 nm, they can be used as a vehicle for systemic administration, thanks to their prolonged blood circulation [54]. Their small size also enables the uptake of polydopamine by cancer cells [55]. Due to its excellent biocompatibility, pH sensitivity, and good adhesion, polydopamine is a suitable material for use in the treatment of cancer. To this day, several materials with polydopamine have been used in head and neck cancers, taking advantage of its many properties (Table 1).

Most oral cancers are derived from epithelial and mucosal mutations found in the exposed parts of the mouth, which enables the use of photothermal therapy [56]. Li et al. developed a pH-responsive charge reversal nanomedicine system for oral cancer. They synthesized polydopamine-modified black phosphorus nanosheets (BP NSs) as basal material, then used polyacrylamide hydrochloride-dimethylmaleic acid (PAH-DMMA) charge reversal system for further surface modification, which can be negatively charged at blood circulation, and become a positive surface charge in the tumor site weakly acidic conditions due to the breaking of dimethylmaleic amide [57]. Polydopamine coating not only enhanced the photothermal properties of this material but also greatly improved its stability [11]. BP@PDA-PAH-DMMA constructed by Li et al. was suitable for intravenous delivery, the ability to promote tumor cell uptake, as well as excellent photothermal properties in vivo and in vitro due to the use of polydopamine, and the killing effect of oral cancer cells, providing a new idea for the treatment of oral cancer [57].

The epidermal growth factor receptor (EGFR) is overexpressed in many neoplastic cells, including head and neck cancer. Cetuximab, a chimeric anti-EGFR monoclonal antibody, has been approved by the FDA as an EGFR inhibitor for the treatment of colorectal cancer and head and neck cancer [58]. He et al. developed a photothermal converting nanomaterial based on the core/shell structure of biodegradable poly(lactide-co-glycolide) (PLGA) and polydopamine and to enhance effectiveness they encapsulated doxorubicin (DOX) into the cetuximab functionalized nanoparticle [59]. With the help of an anti-EGFR antibody, the nanoparticle could efficiently penetrate head and neck cancer cells and convert near-infrared light into heat to trigger drug release from the PLGA core. Scientists used polydopamine because polydopamine nanoparticles can generate heat after NIR irradiation [60]. In the experiment by He et al., it was shown that the PLGAa nanoparticle itself did not generate heat after irradiation, and PLGA/PD showed a concentration-dependent photothermal effect. Due to the unique concentration of PLGA/PD, overheating or burnout situations could be easily prevented. Since the nanoparticle was retained in the tumor tissue and then released, the cardiotoxicity associated with the use of doxorubicin was minimal. The authors stated that thanks to the biodegradability of DOX@PLGA/PD-C, the nanoparticle may be a promising tool in the treatment of head and neck cancer [59].

Maor et al. investigated if NIR laser-induced photothermal response could expedite the release of theranostic agents like copper oxide nanoparticles (CuO-NPs) from PLGA nanospheres, coated with the efficient light-absorbing PDA. Maor et al. demonstrated the significant effect of the PDA shell in heat induction when they irradiated the target with the NIR-laser beam after using polydopamine to enable simultaneous complementary photothermal therapy. Heating efficiency was higher than 85%, compared to uncoated copper oxide nanoparticles loaded with PLGA or water, as a control group [61]. The excellent adhesive properties of polydopamine make it possible to use it to cover the surfaces of polymeric nanocarriers and inorganic NPs. Due to the surface modification, it was possible to obtain a stealth effect using PEG to reduce interaction with the immune system. PEGylation is also commonly used to extend the circulating half-life by functionalizing a hydrophilic polymer such as PEG at the surface of a nanoparticle. In the case of polydopamine, the -SH or -NH2 terminated polyethylene glycol can be modified on the PDA surface, which can further reduce the recognition and destruction of nanoparticles by the reticuloendothelial system (RES), thereby extending the circulation time [23]. Maor et al. expressed that controlled release connected with high heating efficiency allows the designing of a thermal therapy approach, capable of killing tumor cells with lower laser power and shorter time than in conventional therapies. The results also showed the effect of the PDA coating on heat induction and showed that it mainly influences temperature. They attributed the light-induced response to the polydopamine coating as a light-sensitive polymer. Its wide absorption spectrum, especially in the first biological optical window (650–950 nm), allows for an increase in temperature at a level useful in the treatment of hyperthermia [61].

Jin et al. in their studies designed a nanocomplex PDA–SNO–GA–HA–DOX (PSGHD) to enable effective and accurate tumor therapy. PSGHD nanocomplex was tested in vitro with tongue squamous cell carcinoma (HN6 cells) and in vivo of WSU–HN6 tumor-bearing nude mice. Their multi-mode therapy was based on four different functions, whilst polydopamine was used for its photothermal conversion. The coactive effect of photothermal conversion by polydopamine and enzyme-triggered gambogic acid released from a gambogic acid-derivatized hyaluronic acid (HA–GA) resulted in tumor microenvironment-dependent gentle photothermal therapy [62].

New therapeutic strategies for papillary thyroid cancer based on organelle-targeted nanomaterials are very welcome to avoid excessive treatment with conventional surgery. Wang et al. demonstrated a strategy to inhibit mitochondria-targeted strategy and exocytosis inhibition of PDA-coated inorganic nanoparticles to enhance therapy for papillary thyroid cancer. Polydopamine has been used as a universal, multi-functional surfactant for coating inorganic NPs because of its abundant catechols that react with thiols and amines via Michael addition or Schiff base reactions. Taking advantage of its strong photothermal action, PDA-coated nanomaterials exhibit enhanced photothermal performance and act as promising photothermal agents in cancer treatment. They combined the mitochondria-targeted approach with photothermal therapy to achieve a non-invasive and improved thermal ablation of TPC-1 cells. These findings indicated that it has been shown that PDA-coated inorganic NPs can be used to develop a mitochondria-targeted anti-cancer therapy strategy that can selectively release drugs into the mitochondria and support cancer treatment by inhibiting exocytosis [63].

Esophageal cancer is a difficult disease to treat and has a high mortality rate. Doctors used stent implantation as the primary treatment method. However, neoplastic and inflammatory cells severely interfere with the clinical use of the stent and limit its long-term performance. One solution is to provide the stent with a continuous anti-cancer function. For this purpose, Zhang et al. synthesized a functional layer consisting of polydopamine and polyethyleneimine (PEI). As polydopamine has strong binding properties, which makes it possible to conjugate molecules [64]. In this study, polydopamine is used in the PDA/PEI layer to bind polyethyleneimine. It is possible due to the great amount of imino group in PEI, which can be attached via Michael addition and Schiff base reaction. Polyethylenimine was used, due to its good mechanical property and stability, and has also been proven to kill cancer cells [65]. In this work, esophageal cancer cells were cultured on each surface to evaluate the PDA/PEI layers’ anti-cancer function. The results demonstrated the PDA/PEI layers possessed excellent and continuous anti-cancer function, suggesting the promising potential of the layers for the application on surface modification of the esophageal stent materials [66]. Similar use of polydopamine was found in Zhang et al. later research, where they studied the PDA/PEI/5-Fu coatings, which inhibited the esophageal tumor cells (Eca109), epithelial cells (Het-1A), fibroblast (L929) and macrophages adhesion to the surface [67].

Polydopamine, as stated before, has a high photothermal performance, however, it can also be used for bonding PS to obtain photodynamic activity. In previous research, PDA with photosensitizers showed their anticancer effects in vitro and in vivo, however, using them may lead to PS release into blood circulation. To prevent the release of photosensitizers in the blood, it is attached to polydopamine nanoparticles by covalent bonds. However, the cleansing properties of ROS pose problems. Controlled release of PS at the tumor site is possible by combining the photosensitizers with sensitive materials that are cleavable by the relevant tumor microenvironment, such as acidic pH, oxidative stress, or an enzymatic reaction, and also through external energy sources such as light. To prevent this from happening, Zmerli et al. proposed affixing a TS linker to the PS, which would enable PS release in the selected tissue [42]. Zmerli et al. synthesized and characterized a new PEGylated PDA-based nanoplatform with bifunctional PTT and PDT properties, allowing bimodal cancer therapy with the possibility of releasing the photosensitizer on demand by bonding PS to polydopamine by covalent bonds. PEGylation of nanoparticles is a frequently used approach to increase their circulation time in the bloodstream or improve drug delivery efficiencies to target cells and tissues [68]. Created nanoplatforms showed low cytotoxicity in vitro, with high photothermal conversion efficiency and higher photodynamic effects on esophageal cancer cells [42].

At this stage of the polydopamine research, we can say that there are several unresolved problems. Scientists consider polydopamine a biocompatible compound. There are results confirming its biocompatibility with head and neck cancer cells. In the He et al. study, they conducted a live and dead assay study for polydopamine with PLGA on head and neck cancer cells. For cells co-incubated with the PLGA/PD nanoparticle, almost all cells were green, suggesting that the PLGA/PD nanoparticle itself is non-toxic [59]. Studies also show the biocompatibility of polydopamine in vivo [69]. However, with preclinical studies, there are only short-term research and at the moment there are no studies of chronic toxicity. Especially, one based on an animals model which would enable obtaining the results needed for further work on the use of nanostructures based on polydopamine. There are also no studies on the possible accumulation of polydopamine or its biodistribution, and biodistribution is a key issue in future clinical applications of PDA. The nanoparticles have the potential to interact with the mononuclear phagocyte system (MPS), which is made up of lymph nodes, spleen and liver and consists of immune cells responsible for identifying and removing foreign bodies from the blood. In addition, their surface coating may also affect the fate of therapeutic systems [70]. Another important challenge of using PDA is its fate in the body. Antibodies, peptides, and other bio-recognition molecules can often be used to target nanoparticles in nanotherapy systems precisely using PDA in theory, but in practice, nanoparticles undergo non-specific binding and endocytosis in vitro [71] and uptake by RES in vivo [71], which may be a case also with polydopamine. One more important limitation of the use of polydopamine in clinical trials is the difficulty of its structural identification. PDA analysis is further complicated by its insolubility in water as well as in organic solvents. As a result, the PDA study using typical analytical tools, such as e.g., solution copy NMR spectrum, solution UV-vis spectroscopy, gel permeation chromatography and many mass spectroscopy techniques are not possible [72]. All of this makes the understanding of polydopamine in the body incomplete and inhibit the further development of PDA-based nanostructures in forthcoming clinical trials. Future research that would focus on solving these problems would help with the potential future clinical application of these PDA-based nanostructures.

**Table 1 ijms-24-04890-t001:** Use of polydopamine in the treatment of head and neck cancer.

Reference	Nanotherapeutic System	Use of the Nanotherapeutic System	Use of Polydopamine	Results
Li et al., 2021 [57]	Polydopamine-modified black phosphorus nanosheets functionalized with polyacrylamide hydrochloride-dimethylmaleic acid (BP@PDA-PAH-DMMA).	pH-responsive charge reversal nanomedicine system for oral cancer.	Enhancing photothermal properties; improving stability.	Compared with the temperature change of bare BP NSs, the temperature of BP@PDA increased, which was connected to the good photothermal conversion efficiency of the PDA layer.
Maor et al., 2021 [61]	PLGA nanospheres encapsulating copper oxide nanoparticles (CuO-NPs) and coated with PEGylated polydopamine (CuO-NPs@L-PLGA/PDA/ PEG).	Laser-induced thermal response and controlled release of copper oxide nanoparticles from multifunctional polymeric nanocarriers in Cal-33 head and neck squamous cell carcinoma cells.	Photothermal therapy; surface modification.	Results showed that controlled release connected with high heating efficiency allows the designing of a thermal therapy approach, capable of killing tumor cells with lower laser power and shorter time, than in conventional therapies.
Zmerli et al., 2021 [42]	PEGylated PDA-based nanoplatform (PDA−PEG NPs).	As a nanoplatform with multifunctional properties including photothermal and photodynamic activities for esophageal cancer cell lines (KYSE-30).	Photothermal therapy; Photodynamic therapy; surface PEGylation.	They showed very low dark cytotoxicity in vitro, with a simultaneous high photothermal conversion efficiency, along with a higher photodynamic effect on cell lines using active targeting. It was possible to obtain a synergistic phototoxic effect after irradiation at two different wavelengths.
Jin et al., 2020 [73]	S-nitrosothiol functionalized polydopamine core and a gambogic acid-derivatized hyaluronic acid shell with doxorubicin (PDA–SNO–GA–HA–DOX).	For pH-induced chemotherapy, enzyme-triggered low-temperature photothermal therapy and NIR-dependent gas therapy for HN6 tongue squamous cell carcinoma treatment.	Photothermal therapy; Increasing biocompatibility, biodegradability; Enabling release of DOX in an acidic environment.	Studies have shown that nanocomplex achieved both tumor-specific chemotherapy and low-temperature photothermal therapy due to the use of polydopamine. In vivo studies from mouse models showed that the PSGHD nanocomplex can completely inhibit tumor growth and significantly extend the survival of tumor-bearing mice for 50 days.
Wang et al., 2018 [23]	PDA-coated gold–silver alloy NPs (Au–Ag@PDA NPs).	Mitochondria-targeted and exocytosis inhibition strategy of PDA-coated nanoparticles for enhanced papillary thyroid cancer therapy.	Photothermal therapy; surface modification.	Researchers investigated the endocytosis pathway, subcellular localization, and cellular responses between nanoparticles and TPC-1 cells. The results showed that the nanoparticles are hardly excreted by TPC-1 cells, leading to cell cycle arrest and survival pathway mediated by autophagy. With photothermal therapy, it was possible to obtain non-invasive thermal ablation.
He et al., 2017 [59]	Doxorubicin encapsulated in PLGA/polydopamine nanoparticle and functionalized with anti-EGFR antibody (DOX@PLGA/PD-C).	Photothermal and chemotherapy of UMSCC 22A head and neck cancer cells.	For photothermal properties and enabling surface modification.	Since the PLGA nanoparticle itself could not produce heat upon NIR irradiation, research showed that PLGA/PDA nanoparticle quickly heated the solution temperature to 45 °C, indicating that PLGA/PDA is a good photothermal converting material, preventing overheating or overburn situation. Because the nanoparticle was retained in the tumor tissue and only the sweat released its load inside the cancer cells, it was possible to minimize the cardiotoxicity associated with doxorubicin.
Zhang et al., 2017 [66]	Polydopamine and Poly-ethylenimine layers (PDA/PEI).	Modification of the surface of esophageal stents through the PDA/PEI layer to give them an anti-cancer effect.	For PDA good binding properties, allowing to bind imino groups of PEI via Michael addition and Schiff base reactions.	PDA/PEI layers showed remarkable improvement in esophageal cell apoptosis and necrosis, suggesting excellent anti-cancer function.

## 11. Conclusions

Head and neck cancers represent a wide range of cancers. The prognosis depends on the age, sex, and location of the tumor. In the last few years, interest in the use of polydopamine in head and neck cancers has increased. In this review, we demonstrated the potential of polydopamine in smart drug delivery systems, and polydopamine-based nanomaterials in particular. A characteristic feature that distinguishes polydopamine is its high adhesion capacity. This enables the use of polydopamine as a surface layer on other nanoparticles, which can modify or improve their activities, and attach various compounds to use the therapeutic effect of the current method of treatment, including chemotherapy and radiotherapy. Most research focusing on head and neck cancer therapy uses polydopamine in photothermal therapy due to its excellent light absorption characteristics to selectively over-heat cancer cells. Another important aspect of polydopamine used in HNC was its chemical reactivity. The many functional groups were found in PDA, which were able to react with a broad spectrum of molecules so that the materials achieved desired properties. Polydopamine-based nanomaterials used in head and neck cancer are just becoming popular, so additional research is still very much needed due to the variety of materials used, as well as their applications in cancer therapy. There is also one more important aspect of this variety of materials used in cancer, and that is toxicity, which is often overlooked. Polydopamine is considered being a biocompatible material, however, as of today there is no long-term toxicity research. This review describes nanotherapeutics systems, which are made of many compounds, and these compounds affect their physicochemical properties and the toxicity of these materials that have not been tested.

## Figures and Tables

**Figure 1 ijms-24-04890-f001:**
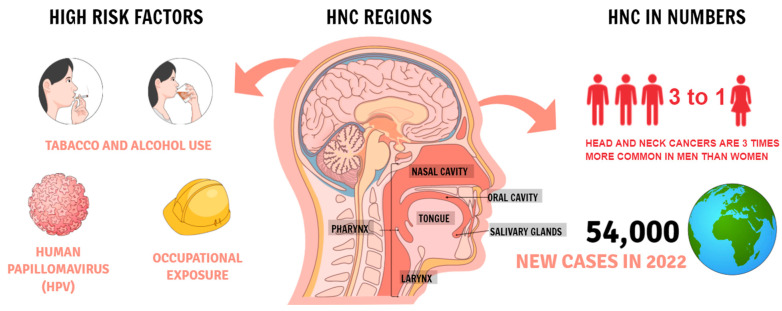
Head and neck cancer statistics according to the SEER.

**Figure 2 ijms-24-04890-f002:**
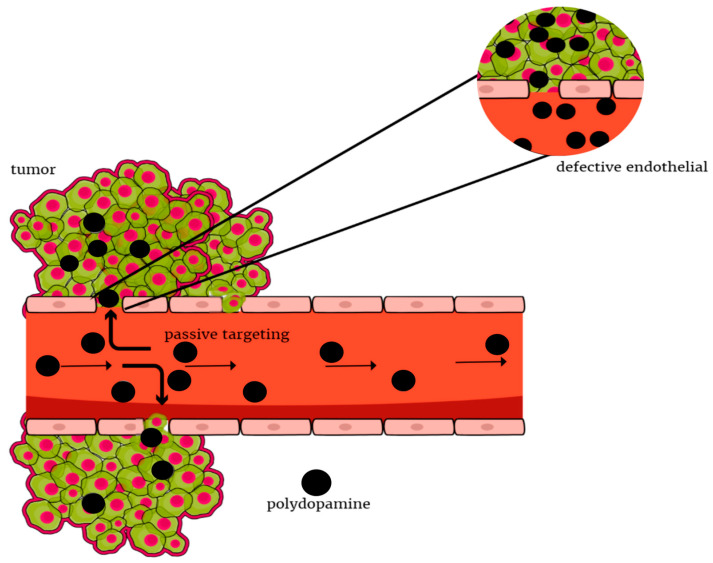
Polydopamine-based materials allowing for passive accumulation of therapeutic agents within tumors via the EPR effect.

**Figure 3 ijms-24-04890-f003:**
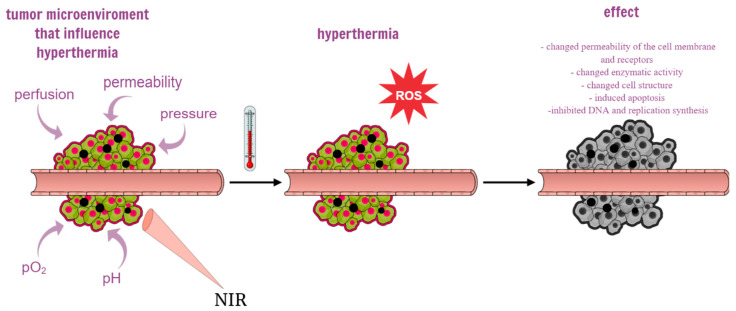
The diagram shows the influence of the tumor microenvironment on hypothermia and the effects of photothermal therapy mediated by polydopamine-based nanoparticles. A large amount of gold nanoparticles accumulates in cancer cells, which causes a photothermal effect in response to near-infrared light (NIR) and reactive oxygen species (ROS), ultimately inducing apoptosis and necrosis of cancer tissue.

## Data Availability

Not applicable.

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
