# Peer review of "Polydopamine-Based Material and Their Potential in Head and Neck Cancer Therapy—Current State of Knowledge"

_ijms, 2023, doi:10.3390/ijms24054890_

Round 1

Reviewer 1 Report

The manuscript is a review of the potential use of polydopamine (PDA) in nanotherapeutic systems for head and neck cancers. The manuscript discusses the limitations of traditional therapies and the potential benefits of using nanomaterials such as PDA, which has unique properties and has found applications in various forms of therapy for HNC. The manuscript appears to be well-written and informative, providing valuable insights into the potential use of PDA in HNC research. However, there are a few areas where the manuscript could be improved.

1. The manuscript could benefit from including more information on the mechanism of action of PDA in HNC therapy, as well as the potential side effects and limitations of using PDA in nanotherapeutic systems.

2. The manuscript could benefit from more in-depth discussions on their efficacy and safety in preclinical and clinical studies. The manuscript could also discuss the potential challenges associated with translating PDA-based therapies from the laboratory to clinical practice.

Reviewer 2 Report

The manuscript is well written.

The text is well structured in terms of the use of polydopamine materials in the treatment of oncological diseases (chemo-, photo-, radiation etc).

Necessary corrections and additions:

1. It would be nice to add one more section:  the chemical structure of polydopamine, a description of the chemical properties and some schematic illustration of the synthesis of PDA materials

2. It is imperative to improve the quality (increase the resolution) of Figures 1 and 2 (including insert).
